# Insight into the Speciation of Heavy Metals in the Contaminated Soil Incubated with Corn Cob-Derived Biochar and Apatite

**DOI:** 10.3390/molecules28052225

**Published:** 2023-02-27

**Authors:** Truong Xuan Vuong, Joseph Stephen, Thi Thu Thuy Nguyen, Viet Cao, Dung Thuy Nguyen Pham

**Affiliations:** 1Faculty of Chemistry, TNU-University of Science, Thai Nguyen City 24000, Vietnam; 2School of Materials Science and Engineering, University of NSW, Kensington, NSW 2052, Australia; 3Institute of Resources, Ecosystem and Environment of Agriculture, Center of Biochar and Green Agriculture, Nanjing Agricultural University, Nanjing 210095, China; 4School of Environmental and Rural Science, University of New England, Armidale, NSW 2351, Australia; 5ISEM and School of Physics, University of Wollongong, Wollongong, NSW 2522, Australia; 6Faculty of Natural Sciences, Hung Vuong University, Viet Tri City 35120, Vietnam; 7NTT Institute of Applied Technology and Sustainable Development, Nguyen Tat Thanh University, Ho Chi Minh City 70000, Vietnam; 8Faculty of Environmental and Food Engineering, Nguyen Tat Thanh University, Ho Chi Minh City 70000, Vietnam

**Keywords:** metal remediation, chemical fraction, heavy metal pollution

## Abstract

Soil heavy metal contamination is a severe issue. The detrimental impact of contaminated heavy metals on the ecosystem depends on the chemical form of heavy metals. Biochar produced at 400 °C (CB400) and 600 °C (CB600) from corn cob was applied to remediate Pb and Zn in contaminated soil. After a one month amendment with biochar (CB400 and CB600) and apatite (AP) with the ratio of 3%, 5%, 10%, and 3:3% and 5:5% of the weight of biochar and apatite, the untreated and treated soil were extracted using Tessier’s sequence extraction procedure. The five chemical fractions of the Tessier procedure were the exchangeable fraction (F1), carbonate fraction (F2), Fe/Mn oxide fraction (F3), organic matter (F4), and residual fraction (F5). The concentration of heavy metals in the five chemical fractions was analyzed using inductively coupled plasma mass spectroscopy (ICP-MS). The results showed that the total concentration of Pb and Zn in the soil was 3023.70 ± 98.60 mg kg^−1^ and 2034.33 ± 35.41 mg kg^−1^, respectively. These figures were 15.12 and 6.78 times higher than the limit standard set by the United States Environmental Protection Agency (U.S. EPA 2010), indicating the high level of contamination of Pb and Zn in the studied soil. The treated soil’s pH, OC, and EC increased significantly compared to the untreated soil (*p* > 0.05). The chemical fraction of Pb and Zn was in the descending sequence of F2 (67%) > F5 (13%) > F1 (10%) > F3 (9%) > F4 (1%) and F2~F3 (28%) > F5 (27%) > F1 (16%) > F4 (0.4%), respectively. The amendment of BC400, BC600, and apatite significantly reduced the exchangeable fraction of Pb and Zn and increased the other stable fractions including F3, F4, and F5, especially at the rate of 10% of biochar and a combination of 5:5% of biochar and apatite. The effects of CB400 and CB600 on the reduction in the exchangeable fraction of Pb and Zn were almost the same (*p* > 0.05). The results showed that CB400, CB600, and the mixture of these biochars with apatite applied at 5% or 10% (*w*/*w*) could immobilize lead and zinc in soil and reduce the threat to the surrounding environment. Therefore, biochar derived from corn cob and apatite could be promising materials for immobilizing heavy metals in multiple-contaminated soil.

## 1. Introduction

Heavy metal contamination has been a severe issue in many countries worldwide [1,2]. Human activities such as farming, mining, industry, and transportation are the primary sources that induce heavy metal pollution in the surrounding ecology [1,3]. Some heavy metals including zinc (Zn), copper (Cu), and manganese (Mn) are essential minerals to humans and plants at a minimal concentration and become toxic when their concentration is elevated [2,4]. At the same time, other HMs such as lead (Pb), cadmium (Cd), arsenic (As), and mercury (Hg) are toxic in trace amounts [2]. Heavy metals (HMs) can cause widespread environmental pollution including soil, sediment, water, and plants [5]. Consequently, this leads to the destruction of the ecology and induces a potential threat to human health via the food chain (soil–plant–human or soil–plant–animal–human) [4], especially to those who are dwelling near mining areas [5]. Many studies have reported that mining activities are one of the most substantial sources of soil heavy metal pollution [6,7]. HMs present in the soil are stable chemicals, since they are non-degradable and persist in the environment; therefore, they have long-term effects on the soil and the ecosystem [1,8].

Many methods, which are based on physical, chemical, and biological processes, have been applied to remediate heavy metals in contaminated soil [2,4]. Chemical processes including chemical stabilization, electrochemical remediation, chemical soil washing, treatment with nanoparticles, and stabilization or solidification are very effective methods for remediating heavy metals in contaminated soil [2]. Chemical stabilization, one of the most popular chemical processes, can decrease the mobility and bioavailability of heavy metals in the soil by adding specific amendments [2]. The most commonly used modifications are various chemicals including limes [9,10,11], phosphate compounds [12,13,14,15], and organic compounds [16,17,18,19]. Nowadays, biochar, rich in carbon with large organic functional groups, is a ubiquitous amendment used in remediating heavy metals in contaminated soil [20,21,22,23]. It has been widely studied in remediating heavy metals in polluted soil since it has specific characteristics such as a somewhat high cation exchange capacity, porous structure, and large surface area [24]. The ability of biochars to adsorb and stabilize heavy metals improves over time by forming organomineral microaggregates [25]. Biochar is primarily produced from various biomass such as agricultural waste (rice straw, sugarcane bagasse) [24] and wood (willow wood, hardwood) [22], which are available and cost-effective materials [26].

Many previous studies have reported that corn cob-derived biochar has the potential to remediate contaminants in polluted soil [27,28,29]. Apatite ore (AP), rich in phosphorous, has been used to remediate heavy metals in soil in many previous studies [13,30,31,32,33]. The combination of biochar and apatite in remediating heavy metals in soil has also reported in some studies [34,35]. However, there is limited information about the effectiveness of the combination of various biochars with apatite in remediating heavy metals. This paper focuses on assisting in filling this information gap.

The present study aimed to study the impact of corn cob-derived biochar and the combination of biochar with apatite on the soil properties and speciation of heavy metals such as Pb and Zn, which dominate in the multi-contaminated farming soil. We hypothesized that biochar derived from corn cob and the mixture of biochar with apatite could transform heavy metals from labile fractions into stable fractions in contaminated soil. Therefore, this study was conducted to ascertain (1) the characteristics of corn cob-derived biochar produced at 400 °C (CB400) and 600 °C (CB600); (2) the alteration of soil properties after being treated with biochar and apatite; and (3) the effects of biochar and the blend of biochar and apatite (AP) to the chemical fractions of Pb and Zn, particularly the exchangeable fraction, for heavy metal remediation.

## 2. Results and Discussion

### 2.1. Characteristics of the Investigated Soil and Amendments

The primary characteristics of the soil, apatite, and biochar are shown in Table 1. The pH value of the studied soil was 6.69, demonstrating that the soil was neutral, according to the FAO’s classification. In addition, the organic carbon (OC) and electrical conductivity (EC) values of the soil were 2.19 (± 0.40)% and 136.51 (± 0.50) µS cm^−1^, respectively. The OC and EC values of the studied soil were much smaller than those of the amendments (CB400, CB600, and AP), especially the EC value (see Table 1). Moreover, the mean concentration values of Pb, Zn, and Cd in the soil were 3023.70 ± 98.60 mg kg^−1^, 2034.33 ± 35.41 mg kg^−1^, and 14.11 ± 0.93 mg kg^−1^, respectively. This result was in agreement with previous studies [35,36]. According to the United States Environmental Protection Agency (U.S. EPA 2010), the acceptable standard of Pb, Zn, and Cd in agricultural soil were 200, 300, and 3 mg kg^−1^, respectively. This means that the concentration of Pb, Zn, and Cd in the investigated soil was about 15.12, 6.78, and 4.70 times higher than the allowable limits of Pb, Zn, and Cd set by the U.S. EPA (2010), illustrating that this soil sample was heavily contaminated by heavy metals. Therefore, it is necessary to find a solution to remediate Pb and Zn in contaminated soil. In this study, we concentrated merely on remediating Pb and Zn in the soil due to the extremely high concentration of these contaminated elements.

In contrast to the soil, the amendments had minimal Pb, Zn, and Cd (Table 1), showing that they are suitable materials to amend in soil to remediate heavy metals [35,37]. In addition, CB400, CB600, and AP had pH values of 8.11, 9.71, and 9.16, respectively. The high pH values of CB400 and CB600 were ascribed to the decomposition of acidic organic groups in the raw corn cob at high pyrolyzed temperatures, and the high pH value of AP could be assigned to the alkaline substances in the apatite ore. These values were significantly higher than that of the investigated soil and indicates that after being incubated with these materials, the pH of the incubated soil samples would increase due to the neutralized reaction of alkaline substances in the amendments [38], leading to the potential of immobilizing heavy metals in the soil [37].

### 2.2. Characteristics of Amendments

#### 2.2.1. FTIR Analysis of Amendments

The FTIR results of the pristine corn cob (CC), corn cob-derived biochar produced at 400 °C (CB400), and the corn cob-derived biochar produced at 600 °C (CB600) are shown in Figure 1A, while that of apatite (AP) is shown in Figure 1B. The FTIR spectra of samples CC and CB400 had a strong and broad absorption peak at 3440 cm^−1^, while CC600 had an almost flat absorption peak in this region (Figure 1A). This peak is associated with the O–H group’s valence fluctuations [39], whose content was less at higher pyrolysis temperatures [40]. The higher the pyrolysis temperature, the more the O–H group decomposed, leading to the decrease in the intensity of this peak in the FTIR diagram of CB400 and CB600. In addition, the FTIR spectrum for the absorption peak at 2926 cm^−1^ was assigned to the valence oscillation of the C–H bond [41]. This peak was medium for CC, weak for CB400, and almost insignificant for CB600. The decrease in the intensity of this peak might be attributed to the decomposition of the C–H group at high pyrolyzed temperatures. The CC, CB400, and CB600 spectra had another peak at about 2370 cm^−1^, ascribed to the C≡C elongated alkyne functional group [42]. This peak was most noticeable in CB400, less distinct in CC, and almost insignificant at CB600. The peak at about 1744 cm^−1^ corresponded to the prolonged oscillation of the aromatic C=O functional group, the carboxylic group, or the conjugated ketone [41,43]. The elongated C=O and aromatic C=C vibrations were attributed to the peak at about 1510–1620 cm^−1^ [41,44], while those in the region of 1305–1380 cm^−1^ refer to phenolic –OH and C–O, respectively [37], and these peaks appeared mainly in CC, and were barely present in the CB400 and CB600 samples since they were decomposed at a high temperature. The pronounced peak at about 1035 cm^−1^ was associated with C–O–H or C–O–C stretching [39,45]. This peak appeared prominently in CC, but less in CB400 and CB600. The noticeable peak at ~600–622 cm^−1^, evident in CC and insignificant at CB400 and CB600, was assigned to the C–OH out-of-plane bending mode of aromatic compounds [43]. After heating at 400 °C and 600 °C, the stretching vibrations of –OH, fatty acids C–H and C–O at 3440, 2926, and 1035 cm^−1^ decreased significantly, illustrating the decomposition of the cellulose, hemicellulose, and lignin, respectively [46]. The differences in the FTIR results among CC, CB400, and CB600 were assigned to the decomposition of hemicellulose and cellulose in the pristine corn cob, which was rich in cellulose [47]. Hemicellulose decomposes in the temperature range of 220–315 °C and cellulose pyrolysis takes place at 315–400 °C [47,48]. The difference in CB400 and CB600 in the FTIR results can be attributed to the decomposition of lignin/cellulose-derived transformation products at 400 °C. These substances were decomposed at 400–900 °C [47,49]. Hence, CB600 has quite similar peaks to CB400, however, the intensity was much lower due to the almost decarbonization of lignin [49]. To sum up, the F-IR results showed that the distinctive peaks of CC, CB400, and CB600 were primarily associated with the functional groups (–OH, C=O) in cellulose, hemicellulose, and lignin, respectively [37,50], which are the primary components of corn cob feedstock and the corn cob-derived biochar [51].

The infrared spectra of apatite ore (AP) are shown in Figure 1B, which indicate that the FTIR spectra of AP had a pronounced peak at 3446 cm^−1,^ which was assigned to the vibration of the OH group or the absorbed water [33,52]. There were four distinctive peaks at ~1094, 1049, 575, and 464 cm^−1^, which are the typical bands of PO_4_^3−^ (asymmetric stretching vibration of the P–O bond and the asymmetric bending vibration of O–P–O) [33]. The peak at ~797 cm^−1^ confirmed that the F^-^ ion was a component in the apatite sample [53]. Moreover, the peaks that appeared distinctively at around 1424–1437 cm^−1^, ascribed to the CO_3_^2−^ stretching vibration, confirmed that the apatite sample was a fluor-hydroxide-carbonate-apatite [37]. This FTIR result agrees with previous studies [35,36,37]. In conclusion, the FTIR spectra show that the apatite ore has some predominant functional groups such as PO_4_^3−^, OH^−^, or CO_3_^2−^, which can combine with heavy metals to create insoluble forms through precipitation or exchange reactions [37].

#### 2.2.2. SEM-EDS Analysis of Amendments

The SEM image of pristine corn cob (CC), biochar made of corn cob at 400 °C (CB400), biochar made of corn cob at 600 °C (CB600), and apatite (AP) are shown in Figure 2A–D, respectively. Figure 2A,B, and D shows that the surfaces of CC and AP were almost flat with no hollows, while the surface structure of CB400 was not homogeneous, with many cracks and minor holes. The rough surface structure of CB400 was attributed to the decomposition of organic substances such as cellulose and hemicellulose when CC was pyrolyzed at 400 °C [54].

Figure 2C shows that the surface of CB600 was distinguished from CC and CB400 and had many porous holes that varied in size. The highly porous structure of the CB600 was attributed to the discharge of gases (H_2_, CO, CO_2_, and CH_4_) when corn cob was pyrolyzed at 600 °C [54]. The porous levels of the materials were proven by the BET results. The results of the BET method showed that the surface area of BC400 and AP were 1.48 and 0.49 m^2^ g^−1^, respectively, while the surface area of CB600 was 79.63 m^2^ g^−1^ (see Table 1). The highly porous structure of CB600 might facilitate the absorption of heavy metals by CB600.

In addition, EDS was performed to examine the composition of the amendments’ surfaces. The EDS graphs of CC, CB400, CB600, and AP are shown in Figure 3A–D, which show that the main components of CC, CB400, and CB600 were carbon and oxygen, while Si, Mg, O, and P were the predominant elements in AP. These EDS results of AP and corn cob-derived biochar are consistent with previous studies [33,51,55].

To sum up, the results of FTIR, SEM-EDS, and BET show that the functional groups on the surface of CB400 and AP might facilitate the immobilization of heavy metals through precipitation or exchange reactions [56,57,58], while CB600 might facilitate the adsorption of heavy metals due to its large specific surface area [37,59].

### 2.3. Alteration of OC, pH, and EC after a 30-Day Incubation with Biochar and Apatite

The values of pH, electrical conductivity (EC), and organic carbon (OC) are shown in Table 2. The pH value of the control soil (CS) was 6.69, and the pH values of the amended soil samples gradually increased with the increase in the application rates and were significantly higher compared to that of CS (*p* < 0.05). The reason for the increase in the pH values of the soil samples after being amended with biochar and apatite was associated with the elements presented in BC and AP. These elements exist in carbonate or oxide substances and are alkaline when dissolved in water [60,61]. Furthermore, it was noticeable that CB600 and AP caused the pH values of the incubated soil samples to be slightly higher than that of CB400 at the same incubated rates (Table 2). The reason for the distinction was attributed to the higher pH value of CB600 (9.71 ± 0.01) and AP (9.16 ± 0.01) compared to CB400 (8.11 ± 0.01). This result was in agreement with the findings of previous studies, which reported that the pH values of soil incubated with biochar or apatite increased significantly with the increasing rates [61,62].

The OC value of the CS was 19.46 ± 2.14 g kg^−1^, and the OC values of the amended soil samples differed significantly from the CS (*p* < 0.05) (Table 2). Overall, the OC values of the soil samples incubated with BC400 (CB4:3, CB4:5, CB4:10) were somewhat less than those of the soil samples treated with BC600 (CB6:3, CB6:5, CB6:10). These differences were assigned to the higher percentage of carbon in the CB600 (89.36%) than in CB400 (76.17%). The considerable increase in OC of the amended soil samples with the amended rates was also noted in previous studies [36,37,62,63].

The EC value of CS was 120.10 ± 2.50 µS cm^−1^ and significantly smaller than that of the incubated samples (*p* < 0.05). The EC values of the soils incubated with CB400 were slightly higher than those of the BC600-incubated soils (*p* < 0.05). This was attributed to the higher EC of BC600 than BC400 (see Table 2). The high EC values of the amended soils compared to CS were due to the significant amount of soluble compounds in the ash and minerals (Ca, Mg, K) in the biochar and apatite [61]. This result was consistent with previous studies [64,65] that reported that the soil EC increased with the application ratios of biochar. However, there was a particular case when the EC decreased with the increased rates of biochar [62]. In other words, the results depend significantly on the characteristics of the biochar. Awad et al. [62] reported that the EC values of the incubated soils decreased when the incubated proportions of paulownia biochar increased, but increased when the amended ratios of bamboo biochar increased.

In conclusion, after being incubated with biochar CB400, CB600, and apatite, the characteristics of the amended soils changed significantly compared to the control soil (*p* < 0.05). The higher pH values of the amended soils may play an essential role in immobilizing heavy metals via precipitate reactions [66]. Furthermore, the high pH, OC, and EC of the amended soil after being incubated with CB400, CB600, and AP may facilitate the immobility of heavy metals in soils.

### 2.4. Effects of Amendments on the Chemical Fractions of Heavy Metals in the Treated Soil

The chemical fractions of the control soil (CS) and the amended soils are shown in Table 3. Moreover, the proportions of the Pb and Zn chemical fractions are described in Figure 4A,B and Appendix A (see Appendix A). These figures show that the chemical fractions of Pb in the control soil were in the descending sequence of the carbonate fraction (F2: 67%) > residue fraction (F5: 13%) > exchangeable fraction (F1: 10%) > Fe/Mn oxide fraction (F3: 9%) > organic matter fraction (F4: 1%), while the chemical fractions of Zn were in the order of F2~F3 (28%) > F5 (27%) > F1 (16%) > F4 (0.4%). After being incubated with CB400, CB600, and AP at various ratios, the chemical fractions of Pb and Zn had diverse alterations.

#### 2.4.1. Speciation of Lead

Figure 4A,B shows that chemical fractions of Pb in the control soil were in the descending sequence of F2 (67%) > F5 (13%) > F1 (10%) > F3 (9%) > F4 (1%). After being incubated with CB400, CB600, and AP at various ratios, the chemical fractions of Pb in the amended soils had diverse alterations.

The exchangeable fraction (F1): The F1-Pb value of the untreated soil (CS) was 344.90 ± 11.46 mg kg^−1^, and the F1-Pb values of the treated soils decreased gradually with the increasing rates of amendments (Table 3). The F1-Pb values of all amended soil samples were significantly less than that of the control soil (*p* < 0.05), except for sample CB4A3 (332.41 ± 10.19 mg kg^−1^). The effects of amendments on the Pb’s exchangeable fraction were most noticeable in the CB4:10 (245.00 ± 8.10 mg kg^−1^), CB4A5 (245.37 ± 9.06 mg kg^−1^) and CB6:10 (245.48 ± 6.58 mg kg^−1^) samples, where the F1-Pb values decreased by about 28% compared to that of the control soil. Furthermore, the effects of two different biochars (CB400 and CB600) on the exchangeable fraction of Pb were almost the same at various applied rates (*p* > 0.05). After being incubated with biochar and apatite at the rate of 3:3% and 5:5 % (*w*/*w*), there was a slightly different effect on the F1-Pb values of the treated soil samples between CB400 and CB600 (*p* < 0.05). The F1-Pb values of samples CB4A3 and CB4A5 were 332.41 ± 10.19 and 245.37 ± 9.06 mg kg^−1^, respectively, while those of CB6A3 and CB6A5 were 313.14 ± 6.43 and 275.62 ± 9.18 mg kg^−1^, respectively. The reduction in the exchangeable fraction of Pb can be attributed to the ion exchange and precipitation reactions when the pH and the EC of the incubated soil increased after being incubated with biochar and apatite. Cao et al. [67] reported that Pb reacted with other ions in the soil solution to form insoluble substances such as Pb_9_(PO_4_)_6_ or Pb_3_(CO_3_)_2_(OH)_2_ or Pb_5_(PO_4_)_3_OH, leading to the reduction in F1-Pb.

The carbonate fraction (F2): The carbonate fraction of Pb (F2-Pb) in the untreated and treated soils ranged from 62% to 67%, dominating over other fractions (Figure 4A). The domination of this fraction in the investigated soil was also reported in previous studies [36,68] and was associated with the nature of the soil of the studied area [68]. Table 3 shows that the F2-Pb values were slightly reduced when the amended ratios increased, but there were no significant differences between the control soil and treated soils (*p* > 0.05), apart from sample CB6A5 (*p* < 0.05). This result indicates that the amendment of biochar and apatite had no significant effect on the carbonate fraction of Pb when incubated into the soil with ratios of 3, 5, and 10% in one month.

The Fe/Mn oxide fraction (F3): In contrast, the Fe/Mn oxide fraction of Pb (F3-Pb) significantly changed in the incubated soils compared with the untreated soil (*p* < 0.05). The F3-Pb values of the incubated soil samples increased when the amended proportions rose, except for sample CB4A5 (*p* > 0.05). Interestingly, the soil samples incubated with CB600 had F3-Pb values slightly higher than those of the CB400-incubated soil samples (*p* < 0.05), indicating that the CB600 biochar caused the increase in the Pb content in fraction F3 more efficiently than the CB400 biochar after being applied in the contaminated soil, and might be due to the higher values of pH and EC of CB600 compared to those of CB400.

The increase in the F3-Pb values with the increase in the biochar rates was also reported in previous studies [36,62], which reported that applying biochars derived from agricultural wastes such as bamboo and rice straw could significantly increase the Fe/Mn oxide fraction of Pb when the biochar-amended ratios increased, especially at the rate of 5% and 6%. This result might be attributed to Pb’s great affinity to bind to Fe/Mn oxides [62,69].

The carbonate fraction (F2): The carbonate fraction of Pb (F2-Pb) in the untreated and treated soils ranged from 62% to 67%, dominating over other fractions (Figure 4A). The domination of this fraction in the investigated soil was also reported in previous studies [36,68] and was associated with the nature of the soil of the studied area [68]. Table 3 shows that the F2-Pb values were slightly reduced when the amended ratios increased, but there were no significant differences between the control soil and treated soils (*p* > 0.05), apart from sample CB6A5 (*p* < 0.05). This result indicates that the amendment of biochar and apatite had no significant effect on the carbonate fraction of Pb when incubated into the soil with ratios of 3, 5, and 10% in one month.

The Fe/Mn oxide fraction (F3): In contrast, the Fe/Mn oxide fraction of Pb (F3-Pb) significantly changed in the incubated soils compared with the untreated soil (*p* < 0.05). The F3-Pb values of the incubated soil samples increased when the amended proportions rose, except for sample CB4A5 (*p* > 0.05). Interestingly, the soil samples incubated with CB600 had F3-Pb values slightly higher than those of the CB400-incubated soil samples (*p* < 0.05), indicating that the CB600 biochar caused the increase in the Pb content in fraction F3 more efficiently than the CB400 biochar after being applied in the contaminated soil, and may be due to the higher values of pH and EC of CB600 compared to those of CB400.

The increase in the F3-Pb values with the increase in biochar rates was also reported in previous studies [36,62], which indicate that applying biochars derived from agricultural wastes such as bamboo and rice straw could significantly increase the Fe/Mn fraction of Pb when the biochar-amended ratios increased, especially at the rate of 5% and 6%. This result might be attributed to Pb’s great affinity to bind to Fe/Mn oxides [62,69].

The organic carbon fraction (F4): Table 3 shows that the organic carbon fraction of Pb (F4-Pb) in the untreated soil (CS) was 18.41 ± 0.70 mg kg^−1^ and the F4-Pb values in the treated soil increased gradually and significantly in comparison to sample CS (*p* < 0.05). The more the amended ratio increased, the higher the F4-Pb values in the treated soils. The most significant increase in F4-Pb in the soil was in sample BC4:10 when the applied proportion of biochar was 10%. Interestingly, the effects of CB400 were stronger than CB600 in increasing the concentration of Pb in the organic fraction. This result was in agreement with the results reported by previous studies [36,62]. The more substantial effects of CB400 on the organic fraction than CB600 might be due to the higher content of oxygen-containing groups in CB400 compared to CB600, especially the carboxyl, hydroxyl, and phenolic functional groups that facilitate the complex reaction of Pb^2+^ with organic matters [62,70].

The residual fraction (F5): Table 3 and Figure 4A showed that the concentration of Pb in the residual fraction (F5-Pb) in the control soil (CS) was 404.11 ± 11.07 mg kg^−1^ and contributed about 13% of all the fractions, whereas the treated soil samples had F5-Pb values ranging from 402.43 ± 2.11 mg kg^−1^ to 576.34 ± 8.52 mg kg^−1^, contributing from 13% to 19% of all fractions. The F5-Pb values in the treated soils increased significantly compared to sample CS (*p* < 0.05), except for samples CB4A3 and CB6:3, which had no significant difference in comparison with CS (*p* > 0.05). This result was not consistent with the results reported by [36,62,71], which indicate that there was no significant change in the concentration of Pb in the residual fraction when the applied biochars derived from rice straw, paulownia biochar, and bamboo biochar with the ratios of 2, 3, 4, and 6%. However, this result was in agreement with the previous study [72], which reported a significant change in the residual fraction of Pb compared to the control soil when applied peanut shell biochar and wheat straw biochar with a ratio of 5%. The difference in the increase in Pb concentration in the residue fraction could be attributed to the variety of soil types, kind of biochar, incubation time, and pyrolysis temperatures [62,73].

#### 2.4.2. Speciation of Zinc

The speciation of Zn is shown in Table 3 and the proportion of chemical fractions of Zn are shown in Figure 4B and Appendix A (see Appendix A). Figure 4B shows that the chemical fractions of Zn were in the order of F2~F3 (28%) > F5 (27%) > F1 (16%) > F4 (0.4%).

Exchangeable fraction (F1): The concentration of Zn in the exchangeable fraction (F1-Zn) in the control soil (CS) was 361.61 ± 7.98 mg kg^−1^. This figure in the amended soil ranged from 248.69 ± 18.21 to 340.98 ± 17.73 mg kg^−1^ and was significantly reduced in comparison to that of CS (*p* < 0.05), apart from sample CB6A3, which showed no significant difference in F1-Pb compared to CS (*p* > 0.05). The sample with the lowest F1-Zn was CB4A5, which was amended with 5:5% CB400 and apatite (248.69 ± 18.21 mg kg^−1^). The incubation of CB400 and AP reduced the F1-Zn value in CB4A5 by 31.23% compared to the control soil (CS). This result was compatible with previous studies [63,64], which reported a significant reduction in F1-Zn in the soil treated with biochar derived from agricultural waste.

The effects of CB400 and CB600 in reducing F1-Zn were almost the same (*p* > 0.05). They only showed significantly different impacts on F1-Zn when applied together with apatite at the 3% and 5% ratios (*p* < 0.05). In these cases, CB400 had a slightly stronger impact on F1-Zn than CB600 at both ratios (Table 3). Similarly to Pb, the significant reduction in Zn in the exchangeable fraction was ascribed to the high pH and EC of biochar and apatite, leading to the immobilization of Pb or Zn by the cation exchange and precipitate reactions [37,62].

Carbonate fraction (F2): Table 3 shows that the concentration of Zn in the carbonate fraction (F2-Zn) was 615.30 ± 32.60 mg kg^−1^ and accounted for about one-third of all fractions. After being incubated for one month with biochar (CB400, CB600) and apatite, the F2-Zn value increased significantly with the rising application rates compared to the control soil (CS) (*p* < 0.05). The finding was in agreement with previous studies [35,36] that reported a slight escalation in F2-Zn in the treated soils compared to the untreated soil when incubated with rice straw biochar. However, other studies also reported no significant change in F2-Zn between the untreated and treated soil when peanut shell biochar was applied [37]. The disagreement can be assigned to the different kinds of biochar. Furthermore, there was almost no significant difference in F2-Zn between the CB400-incubated soils and CB600-incubated soils at the same amended rate (*p* > 0.05).

Fe/Mn oxide fraction (F3): The content of Zn in the Fe/Mn oxide fraction (F3-Zn) was 622.71 ± 14.70 mg kg^−1^. This figure decreased significantly when CB400 and a combination of CB400 was applied compared to the untreated soil (CS) (*p* < 0.05), while remained unchanged or somewhat decreased when treated with CB600 and a mixture of CB600 and apatite, except for sample CB6A3, which had a higher value of F3-Zn compared to that of CS.

Organic carbon fraction (F4): The concentration of Zn in the organic carbon fraction (F4-Zn) accounted for about 9% of all fractions in the control soil (CS). Similar to Pb, there was a significant rise in F4-Zn in the treated soil when biochar and apatite were applied. F4-Zn increased with the increasing application rates, which is consistent with previous studies [35,36,37]. Furthermore, this value for F4-Zn in the CB400-incubated soil samples was higher than that of the CB600-incubated soil (*p* < 0.05), indicating a more substantial impact of the CB400 biochar in escalating F4-Zn in the treated soil compared to that of CB600 biochar. This finding can be ascribed to the higher content of elements and organic functional groups in CB400 than in CB600, which facilitates the complexation of zinc ions with organic matter [62].

Residual fraction (F5): The content of Zn in the residual fraction (F5-Zn) in the control soil was 591.30 ± 1.65 mg kg^−1^ (Table 3) and accounted for 27% of all fractions (Figure 4B). F5-Zn in the treated soil rose significantly when soil was amended with biochar (CB400 and CB600) and apatite (*p* < 0.05), except for sample CB6A3, which was less than CS. The figure for F5-Zn in the CB400-treated soil was greater than that of the CB600-treated soil at the same amended proportion (*p* < 0.05), illustrating the greater impact of CB400 in converting Zn into a more stable fraction in treated soil than CB600. The finding was attributed to the chemical characteristic of CB400, which had more available elements and active organic groups such as O–H and C=O than in CB600.

Overall, after being incubated with biochar and apatite for one month, there was a significant change in the chemical fractions of the treated soil compared to the untreated soil. The most pronounced alteration of Pb and Zn was in the exchangeable fraction (F1), which had a substantial reduction in the concentration of Pb and Zn, demonstrating the potential of biochar and apatite to immobilize Pb and Zn in the soil, especially at the application rates of 5 and 10%. In addition, the amendment of biochar and apatite also caused an increase in the fraction of F3, F4, and F5 for lead and F2, F4, and F5 for zinc, indicating that biochar and apatite contributed to the conversion of heavy metal from the labile fraction into more stable fractions. These stable fractions, particularly the residue fraction (F5), had a less negative impact on the surrounding environment than the exchangeable fraction under natural conditions.

### 2.5. Mechanism for Immobilizing Heavy Metals

The primary mechanism for immobilizing heavy metals in the soil of amendments such as biochar and apatite, is still an open question since it is usually a combination of various types and depends on the kinds of heavy metals in the soil solution. Many studies have reported the mechanism of the heavy metal remediation of biochar in soil, including chemical and physical adsorption [74,75,76], complexation with active groups [17], cation exchange [24,77], and precipitation with phosphate ions or the hydroxyl ions of an alkaline solution [78,79,80]. In this study, after incubating the biochar and apatite with soil, during the incubation time of 1–4 weeks, soluble cations (Ca, Mg, Al, Si, Fe, Na), anions (Cl, SO_4_, CO_3_, and PO_4_), and organic compounds will be leached from the biochars and apatite particles, and tiny mineral particles (especially Fe/Mn/O) and biochar fragments can break off due to redox reactions [36]. As a result, the pH and EC of the soils around these particles will increase, and there can be reactions with the heavy metals [36]. Previous studies have reported that the main reason contributing to the significant reduction in the exchangeable fraction of heavy metals is the increase in the pH and EC of the soil solution after the application of high pH amendments [62,71,81]. The higher the application ratios, the higher the pH and EC of the soil solution. When the soil pH increased and turned alkaline, the exchangeable fraction of heavy metals was most affected by the soil reaction [74,82]. In the present study, all amendments (CB400, CB600, and AP) had high pH values, which facilitated the hydroxide precipitate reaction; therefore, these factors might significantly contribute to immobilizing heavy metals in contaminated soil through exchange and precipitation reactions. Furthermore, after being incubated with biochar, the pH and EC of the treated soil increased, facilitating the formation of insoluble substances such as Pb_3_(CO_3_)_2_(OH)_2_, and Zn(OH)HCO_3_ [24,76]. The organic functional groups that were available on the surface of CB400 may also play an important role in the exchange and complexation reactions with heavy metals. The high content of organic functional groups of CB400 was confirmed by the FTIR results (see Figure 1A). In contrast, CB600 was poor in organic functional groups (see Figure 1B), but had a high pH value similar to CB400 and a much larger surface area than CB400. Since there was almost no difference between CB400 and CB600 at the same applied ratios, this might illustrate that the adsorption of heavy metals through the large and porous surface of CB600 might play a vital role in immobilizing the heavy metals of CB600.

Additionally, apatite is an ore that is rich in P and other inorganic elements approved by the FTIR and EDS results, facilitating the immobilization of Pb^2+^ via the precipitation of Pb^2+^ with phosphate by creating Pb_3_(CO_3_)_2_(OH)_2_ or Pb_5_(PO_4_)_3_OH [81,83]. As a result, these processes caused the immobility of heavy metals in the soil and turned them into more stable fractions, indicating the possibility of using biochar and apatite in remediating heavy metals such as lead and zinc in the contaminated soil. The formation of stable fractions can be explained by the fact that after 2–4 weeks, an organo-mineral layer formed on the surface of the biochar [36], and this layer consisted of micro-agglomerates made up of nanoparticles of minerals and inorganic compounds that are bonded together by organic compounds, some of which are humic substances [84]. These micro-agglomerates form micro aggregates, which could be incorporated into the stable micro aggregate fraction of the soil when the biochar is fractured [36,85].

In conclusion, we speculate that the main mechanism consisted of various processes such as precipitation, exchange, complexation reactions, and adsorption.

### 2.6. Statistics

#### 2.6.1. Correlation of the Exchangeable Fraction of Pb and Zn with Soil Properties (pH, OC, and EC)

Pearson correlation was performed to investigate the correlation of the exchangeable fraction of Pb (F1-Pb) and Zn (F1-Zn) with soil properties such as pH, OC, and EC. The correlation results are shown in Figure 5A,B. Figure 5A shows that F1-Pb had a moderately negative association with pH (r = −0.53) and a strongly negative association with OC (r = −0.67) and EC (r = −0.70). In the meantime, the correlation of pH with OC and EC was strongly positive (r = +0.65 and r = +0.74), and OC had a very strong positive correlation with EC (r = +0.97) (Figure 5A).

Likewise, the exchangeable fraction of Zn (F1-Zn) had a moderately negative correlation with pH (r = −0.49) and OC (r = −0.57), and a strong negative association with EC (r = −0.68) (Figure 5B), while pH had a strongly positive relationship with OC (r = +0.65) and EC (r = +0.74) In addition, EC had a very close relationship with OC (r = +0.97). The moderate negative correlation of the exchangeable fraction of heavy Pb and Zn has also been reported in previous studies [35,36] that used biochar derived from rice straw and fly ash to remediate heavy metals in multiple-contaminated soil.

The positive correlation of pH, EC, and OC can be attributed to the high value of amendments, especially biochar. CB400 and CB600 were rich in OC and had a high value of EC and pH compared to the control soil. Therefore, the pH, OC, and EC of the incubated soil increased with the rising application rate. The more amendments were applied, the more the pH, EC, and OC were elevated. Consequently, the pH, EC, and OC had a strong positive association in the incubated soil.

In addition, the negative association of the exchangeable of Pb and Zn with the pH, OC, and EC can also be assigned to high values of the pH, EC, and OC of amendments. The pH, EC, and OC in the treated soil rose with the increasing treated ratios of amendments, leading to the increase in the chemical reactions to form insoluble substances via precipitation reactions, cation exchanges, adsorption, and complexation [24,77]. Consequently, the exchangeable fraction of Pb and Zn reduced with the increase in pH, EC, and OC of the soil solution.

#### 2.6.2. PCA Analysis of the Chemical Fractions of Zinc and Lead with pH, OC, and EC

The principal component analysis was also performed to ascertain the association of the Pb and Zn chemical fractions with the soil properties (pH, EC, and OC). The PCA analysis results are described in Table 4 and Figure 6A,B for Pb and Zn.

For Pb, the first principal component (PC1) contributed 58.41% of the total variance, while the second principal component (CP2) accounted for 18.92% of the variance (Table 4). The pH, EC, OC, F3, and F5 were the main positive contributors to PC1, with the r = +0.42, +0.43, +0.45, and +0.35 (Table 4), respectively. In contrast, the exchangeable fraction (F1) and carbonate fraction (F2) of Pb contributed negatively to PC1 with r = −0.42 and −0.24, respectively (Table 4). These results indicate that the pH, EC, OC, F3, and F5 of Pb had a positive association, while the F1 and F2 negatively correlated with the former factors. This finding was consistent with the findings of the Pearson correlation results above-mentioned. In addition, F2 and F4 positively contributed to PC2 (r = +0.52, +0.47, respectively), while the pH, F1, and F3 were negatively contributed to PC2 (r = −0.27, −0.26, and −0.55, respectively). This finding indicates that there might be a factor that caused the strong correlation between F2 and F4 in the treated soil. In addition, the soil samples that had almost no or a weak impact on the chemical fraction change when being incubated with amendments were grouped to create a similar soil group.

Figure 6A indicates that CB6:10, CB6A5, and CB6:5 were in the first group (group 1), and CB4:10 and CB4A5 were in the second group (group 2). These groups significantly changed the Pb chemical fractions compared to the control soil (CS). At the same time, the rest samples (CS, CB6:3, CB4A3, CB4:3, CB4:5, CB6A3) were in the same group (group 3), which had minimal or no effects on the chemical fraction of Pb after being incubated with biochar and apatite.

For Zn, PC1 and PC2 explained 44.99% and 27.16% of the variance, respectively (Table 4). The pH, EC, OC, F2, F4, and F5 contributed positively to PC1, with r = +0.23, +0.42, +0.44, +0.23, +0.37, and +0.37, respectively. In contrast, the exchangeable fraction (F1) and Fe/Mn oxide fraction (F3) of Zn contributed negatively to PC1 with r = −0.41 and −0.23, respectively. These results indicate that the pH, EC, OC, F2, F4, and F5 of Zn had a moderately positive association, while F1 and F3 were negatively correlated with the other factors. The negative association of F1 with other factors such as the pH, OC, and EC in the PCA analysis results was consistent with the correlation result. Moreover, the pH, OC, EC, and F3 contributed positively and primarily to PC2 (r = +0.50, 0.32, 0.30, and 0.54), while F4 and F5 contributed negatively to PC2 (r = −0.37 and −0.32). Likewise, the treated soil samples had almost no or weak impact on the chemical fraction change when incubated with amendments, gathering to form a similar soil group. Figure 6B indicates that CB4:5, CB4A3, CB4A5, and CB4:10 were in the first group (group 1), and CB6:5, CB6A5, and CB6:10 were in the second group (group 2). These groups significantly altered the chemical fractions of Zn compared to the control soil (CS). In comparison, the rest of the samples (CS, CB6:3, CB4A3, CB4:3, CB4:5, CB6A3) were in the same group (group 3), which had minimal or no impact on the chemical fraction of Pb after being incubated with biochar and apatite.

Overall, the PCA analysis illustrates that the chemical fractions of Pb and Zn were affected by the incubation of biochar and apatite to a certain degree, especially the Pb and Zn exchangeable fraction. This finding is consistent with the results of the Pearson correlation.

## 3. Materials and Methods

### 3.1. Sample Collection and Amendment Preparation

The investigated soil samples, surface soil samples, were collected from a corn field in the vicinity of a Pb/Zn mine in Thai Nguyen Province (21°43′46.27″ N, 105°51′2.75″ E), in northern Vietnam. Each soil sample weighs approximately 2 kg and is located about five meters from each other (with a size of roughly 0–20 cm depth, 30 cm width, and 30 cm length). Five subsamples were collected and mingled thoroughly to have a uniform representative soil sample. Biochar was manufactured by cleaning the corn cob with de-ionized water and left to air-dry for 72 h, followed by combustion in a drum pyrolyzer for 2 h at 400 °C (CB400) and 600 °C (CB600) [36,37]. Apatite ore was obtained from Vietnam Apatite Ltd., in Lao Cai Province, Vietnam (22°29′8.02″ N, 103°58′14.38″ E) [37]. The absorbents (biochar and apatite) were pulverized to a size that was less than 1 mm before mingling with the soil [35,37].

### 3.2. Experimental Design

The soil sample was mixed with biochar and apatite at different proportions. Briefly, 100 g of soil was transferred into a plastic cup and then mingled with biochar and apatite at a ratio of 3%, 5%, and 10% in mass (*w*/*w*), respectively. The experiment was set up and described as shown in detail in Table 5. All incubated soil samples were placed in the shadow at ambient temperature for 30 days. During the incubation, deionized water was supplied every two days to retain the soil’s moisture at around 70% [36,37]. After 30 days of incubation, the soil samples were dried at 45 °C for three days and ground to pass through a 2-mm sieve for further analysis [37].

### 3.3. Analysis Methods of Soil and Amendment

#### 3.3.1. Physicochemical Analysis

The basic physicochemical properties of the soil samples and amendments (biochar and apatite) were analyzed before and after one month of being incubated with biochar and apatite. The pH and EC values of the soil and amendments (CB400, CB600, and AP) were measured by a Hanna HI 9124 pH meter with a mixture of samples and deionized water at a ratio of 1:10 (*w*/*v*) of water and substances [86,87]. The pipette method was performed to analyze the soil texture (the fraction of clay, silt, and sand) of the studied soil [88]. Organic carbon (OC) contents in the soil and amendments (biochar and apatite) were analyzed using the Walkley–Black titration method [89].

#### 3.3.2. Heavy Metal Analysis

The soil sample was first treated with a mixture of acid in a microwave oven to analyze the total concentration of heavy metals. In brief, 0.1000 g of the soil sample was scaled and blended with 8 mL concentrated HNO_3_:HCl (*v*/*v* = 1:3) and digested in a Mars 6 microwave system [37,68]. The operating parameters of the microwave system are shown in Appendix A (see Appendix A). The operational parameters of ICP-MS (Agilent 7900) are shown in Appendix A. The recovery of Pb, Zn, and Cd was evaluated by analyzing the sediment standard reference material (MESS-4) and the recovery results of those metals in MESS-4 were 109.27%, 103.22%, and 92.11%, respectively (see Appendix A in Appendix A). The speciation of heavy metals was analyzed using Tessier’s sequence extraction process [90], which included five chemical fractions: exchangeable (F1), carbonate bound (F2), Mn/Fe-hydroxide (F3), organic substance bound (F4), and residue (F5) (see Appendix A Appendix A).

#### 3.3.3. Surface Characteristics of Amendments

Fourier transform infrared spectroscopy (FTIR, JASCO FT/IR- 4600, JASCO International Co. Ltd., Tokyo, Japan) was applied to investigate functional groups on the surface of the apatite and biochar (CB400, CB600) [36,37]. The field emission electron microscope (FE-SEM, JSM-6700F, JEOL, Akishima Tokyo, Japan) equipped with an energy dispersive spectrometer (EDS) was used to analyze the surface morphology and chemical composition of the amendments [91,92]. The surface area and dimensional pore of the materials were investigated using a BET analyzer (TriStar II 3020, Micromeritics Instrument Corporation, 4356 Communications Dr, Norcross, GA 30093, United States) [91,93].

### 3.4. Statistical Analysis

Data analysis was conducted using Excel 2019 and Origin Pro 2021 (OriginLab Corp., Northampton, MA, USA). The mean values of the triplicated results and standard deviation were calculated using Excel 2019. The data in the tables and figures were expressed as the mean value ± standard deviation. One-way ANOVA was performed using Origin Pro 2021 to test the difference in the mean values between the treatments, with *p* < 0.05 considered significant. Principle component analysis (PCA) and Spearman correlation were executed using Origin Pro 2021.

## 4. Conclusions

(i)The physicochemical properties of biochar and apatite were analyzed such as the pH, heavy metal content, and surface characteristics. All amendments including CB400, CB600, and apatite had high pH values and a low content of heavy metals (Pb, Zn, Cd), which are suitable for remediating heavy metals in soils. Additionally, the surface properties of these materials were investigated using FTIR, SEM-EDS, and BET. The FTIR and EDS results showed that CB400 was richer in organic functional groups than CB600, and the apatite was rich in phosphate. At the same time, the SEM and BET results illustrated that CB600 was superior to CB400, and AP in surface area, which might have facilitated the better potential absorption of CB600 than that of CB400 and AP.(ii)After 30 days of incubation, the biochars and apatite positively impacted the soil properties of the treated soil sample such as the pH, OC, and EC. These values of the treated soil samples significantly increased in comparison with those of the untreated samples. The higher the ratio of the amendment incubated, the higher the pH, OC, and EC values, promoting heavy metal remediation in contaminated soil.(iii)After 30 days of incubation, the biochars and apatite had a diverse impact on the Pb and Zn chemical fractions in the treated soil at different application rates. At the same time, the amendments also could increase the F2, F3, F4, and F5 of Pb and Zn by turning them into more stable forms in natural conditions. The main mechanisms are still unknown. However, they might take place via the exchange, precipitation, and complexation reaction of the functional groups and minerals of CB400 and AP as well as the physical adsorption of the large, porous surface of CB600. The most effective application rates of biochar and apatite were 5% and 10% of biochar, while the 3% ratio had no or a slight effect on changing the exchangeable fraction of Pb and Zn. BC400 and CB600 had the same impact on the exchangeable fraction of Pb and Zn when applied at the same ratio of 3, 5, and 10%, indicating that many mechanisms of cation exchange, physical adsorption, precipitation and complexation might occur in contaminated soil when being incubated with CB400 and CB600 as well as the combination of CB400/AP and CB600/AP. This study showed that CB400, CB600, and apatite could be auspicious materials for remediating heavy metals in heavy metal polluted soil.

## Figures and Tables

**Figure 1 molecules-28-02225-f001:**
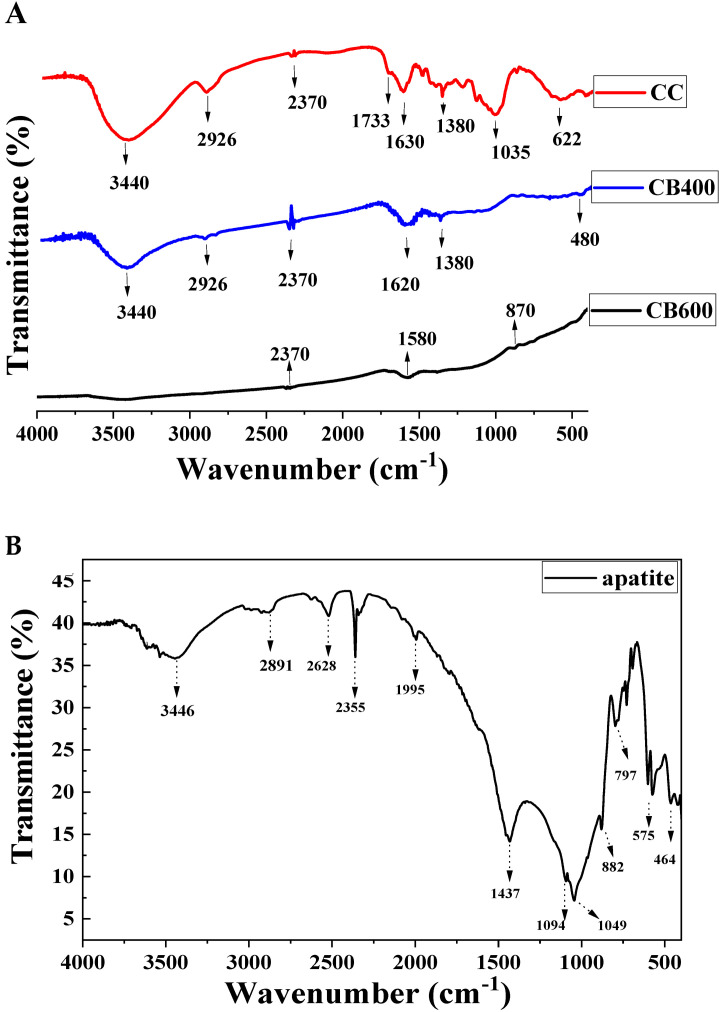
FTIR spectra of corn cob (CC), corn cob-derived biochar at 400 °C (CB400) and 600 °C (CB600) (**A**) and apatite (**B**).

**Figure 2 molecules-28-02225-f002:**
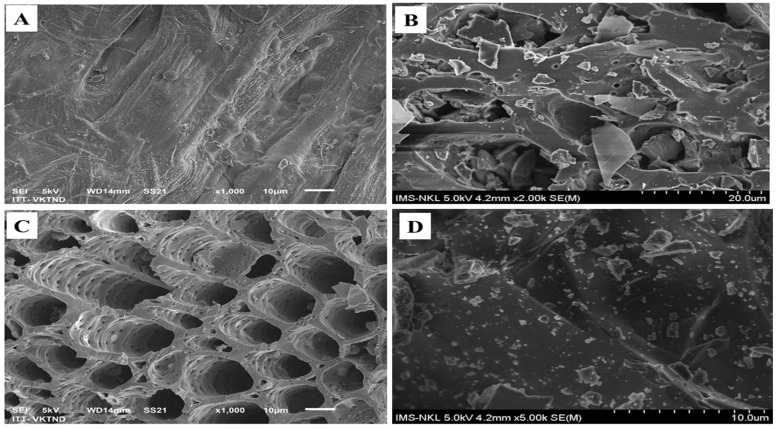
SEM images of CC (**A**), CB400 (**B**), CB600 (**C**), and apatite ore (**D**).

**Figure 3 molecules-28-02225-f003:**
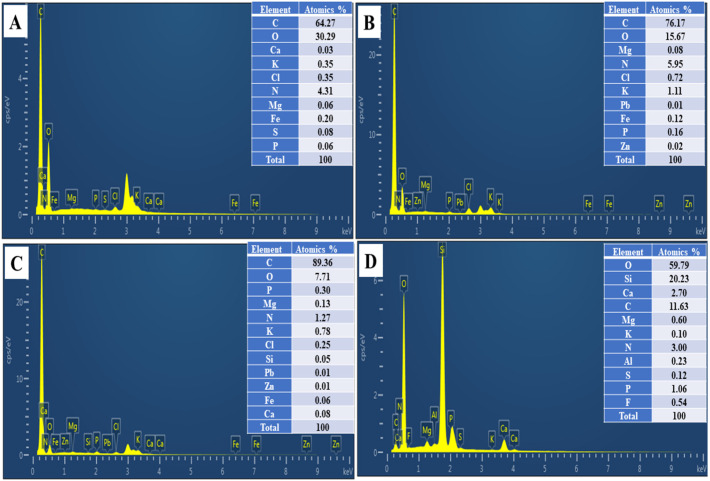
EDS analysis of corn cob (CC) (**A**); corn cob-derived biochar produced at 400 °C (CB400) (**B**), corn cob-derived biochar made at 600 °C (CB600) (**C**), apatite ore (**D**).

**Figure 4 molecules-28-02225-f004:**
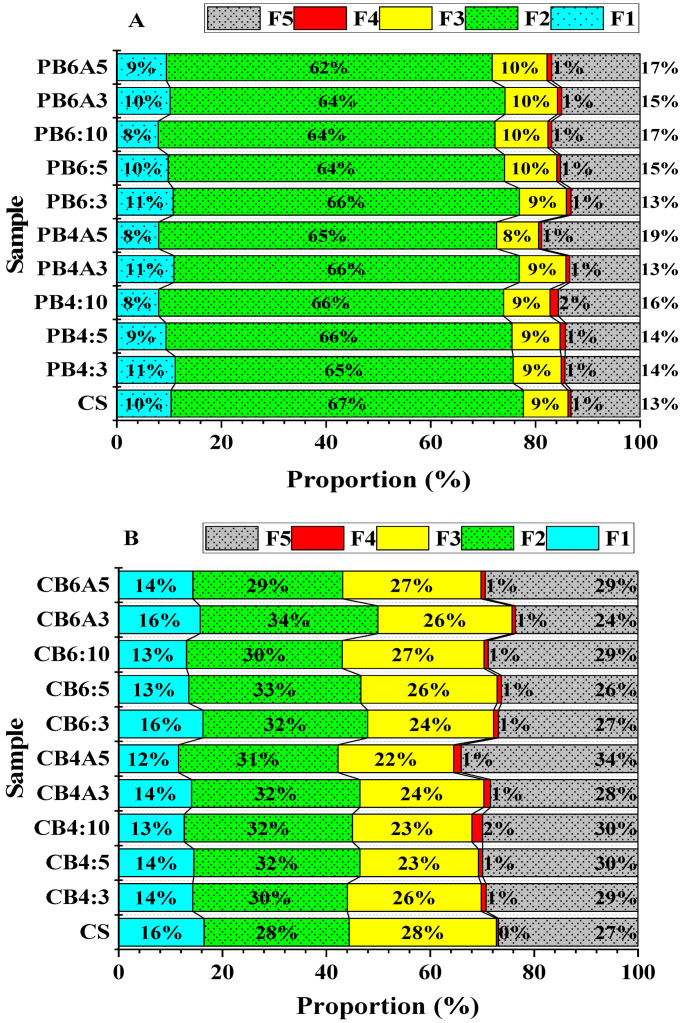
Fractionations of Pb (**A**) and Zn (**B**) in the control soil and amended soil samples.

**Figure 5 molecules-28-02225-f005:**
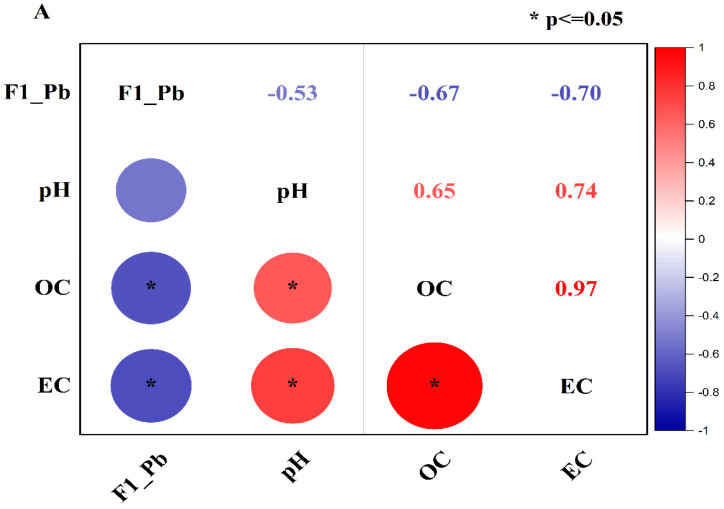
Correlation of pH, OC, and EC with F1, F2, F3, F4, and F5 of Pb (**A**) and Zn (**B**). (* *p* ≤ 0.05 indicates the level of the significant correlation).

**Figure 6 molecules-28-02225-f006:**
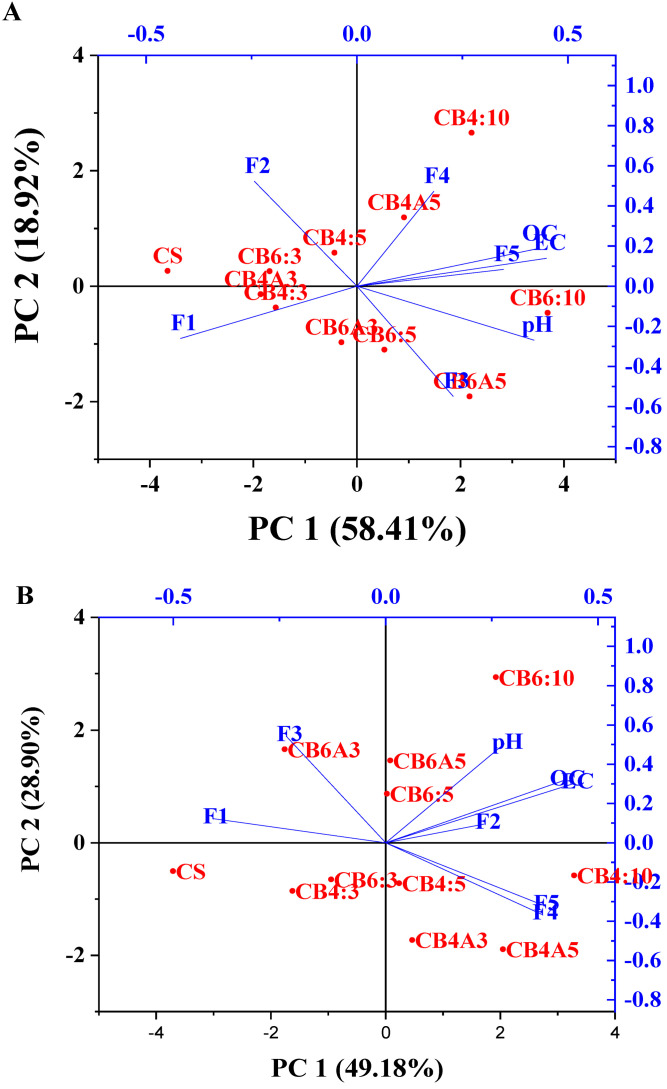
Two-dimensional principal component loading plot in the PCA results of the pH, EC, and OC with chemical fractions of Pb (**A**) and Zn (**B**).

**Table 1 molecules-28-02225-t001:** Physicochemical properties of the soil and amendments [37].

Properties	Unit	Soil	Amendment		
			CB400	CB600	AP
Sand	%	69.78 ± 0.72	-	-	-
Silt	%	5.48 ± 0.32	-	-	-
Clay	%	24.74 ± 0.43	-	-	-
pH		6.69 ± 0.02	8.11 ± 0.01	9.71 ± 0.01	9.16 ± 0.01
OC	%	2.19 ± 0.40	76.86 ± 1.43	84.79 ± 0.95	3.34 ± 0.21
EC	µS cm^−1^	136.51 ± 0.50	2910.01 ± 2.50	3280.02 ± 3.50	1104.50 ± 1.51
Pb	mg kg^−1^	3023.70 ± 98.60	<LOD	<LOD	<LOD
Zn	mg kg^−1^	2034.33 ± 35.41	0.20 ± 0.03	0.50 ± 0.02	9.43 ± 0.03
Cd	mg kg^−1^	14.11 ± 0.93	<LOD	<LOD	<LOD
S_(BET)_	m^2^ g^−1^	-	0.89	79.63	0.49

OC: organic carbon; EC: electrical conductivity; CB400: corn cob-derived biochar produced at 400 °C; CB600: corn cob-derived biochar made at 600 °C; AP: apatite ore; S_(BET)_: surface area, LOD: limit of detection, -: no analysis.

**Table 2 molecules-28-02225-t002:** Exchangeable fraction of Pb (F1-Pb) and Zn (F1-Zn), OC, pH, and EC after a 30-day incubation with biochar and apatite.

Sample	F1-Pb	F1-Zn	pH	OC	EC
	mg kg^−1^	mg kg^−1^		g kg^−1^	µS cm^−1^
CS	344.90 ± 11.46 ^a^	361.61 ± 7.98 ^a^	6.69 ± 0.01 ^h^	19.46 ± 2.14 ^k^	120.10 ± 2.50 ^k^
CB4:3	319.08 ± 2.96 ^b^	330.40 ± 13.81 ^b^	6.93 ± 0.01 ^g^	33.57 ± 1.71 ^h^	194.41 ± 1.50 ^i^
CB4:5	283.64 ± 7.62 ^c^	300.57 ± 18.70 ^d^	6.95 ± 0.01 ^f,g^	39.88 ± 0.96 ^f^	259.50 ± 1.61 ^f^
CB4:10	245.00 ± 8.10 ^d^	291.01 ± 10.58 ^d^	7.01 ± 0.01^e^	83.51 ± 0.80 ^b^	390.10 ± 4.30 ^b^
CB4A3	332.41 ± 10.19 ^a^	311.87 ± 18.15 ^c,d^	7.04 ± 0.01 ^c,d^	34.14 ± 0.96 ^h^	217.80 ± 2.11 ^g^
CB4A5	245.37 ± 9.06 ^d^	248.69 ± 18.21 ^e^	7.06 ± 0.01 ^c^	53.01 ± 1.30 ^c^	308.51 ± 1.51 ^c^
CB6:3	330.72 ± 7.85 ^a^	361.04 ± 17.16 ^a^	6.97 ± 0.01 ^e,f^	34.56 ± 1.45 ^h^	200.30 ± 4.51 ^h^
CB6:5	294.29 ± 8.37 ^c^	298.06 ± 16.39 ^d^	7.03 ± 0.01 ^d,e^	47.46 ± 1.02 ^e^	278.12 ± 2.33 ^e^
CB6:10	245.48 ± 6.58 ^d^	300.58 ± 11.71 ^d^	7.17 ± 0.01 ^a^	90.51 ± 1.09 ^a^	416.51 ± 6.13 ^a^
CB6A3	313.14 ± 6.43 ^b^	340.98 ± 17.73 ^a,b^	7.06 ± 0.01 ^c^	35.59 ± 0.81 ^g^	214.23 ± 3.11 ^g^
CB6A5	275.62 ± 9.18 ^c^	329.03 ± 14.07 ^b,c^	7.11 ± 0.01 ^b^	50.01 ± 1.10 ^d^	297.03 ± 1.54 ^d^

Values = mean ± standard deviation (*n* = 3); Different lowercase letters (a,b,c,d,e,f,g,h,i) in the same column show significant differences (*p* < 0.05) between treatments.

**Table 3 molecules-28-02225-t003:** Chemical fractions of Pb and Zn in the soil incubated with biochar and apatite after 30 days.

Metal	Sample	F1	F2	F3	F4	F5
		(mg kg^−1^)				
Pb	CS	344.90 ± 11.46 ^a^	2063.70 ± 36.31 ^a^	260.51 ± 5.40 ^d^	18.41 ± 0.70 ^f^	404.11 ± 11.07 ^g,h^
	CB4:3	319.08 ± 2.96 ^b^	1994.30 ± 75.90 ^a^	282.60 ± 8.51 ^b,c^	20.12 ± 0.91 ^e^	442.62 ± 5.44 ^e^
	CB4:5	283.64 ± 7.62 ^c^	1985.31 ± 92.30 ^a^	277.43 ± 8.82 ^c^	30.01 ± 1.44 ^b^	428.71 ± 3.65 ^f^
	CB4:10	245.00 ± 8.10 ^d^	2020.50 ± 67.20 ^a^	273.72 ± 9.13 ^c^	49.92 ± 0.53 ^a^	477.42 ± 4.30 ^d^
	CB4A3	332.41 ± 10.19 ^a^	2011.70 ± 56.50 ^a^	272.24 ± 11.94 ^c^	19.91 ± 0.34 ^e^	411.41 ± 5.67 ^g^
	CB4A5	245.37 ± 9.06 ^d^	1975.01 ± 83.20 ^a^	244.72 ± 1.41 ^e^	19.62 ± 0.63 ^e,f^	576.34 ± 8.52 ^a^
	CB6:3	330.72 ± 7.85 ^a^	2023.31 ± 61.50 ^a^	273.71 ± 11.60 ^b,c^	27.61 ± 1.11 ^c^	402.43 ± 2.11 ^h^
	CB6:5	294.29 ± 8.37 ^c^	1924.30 ± 81.50 ^a^	298.80 ± 5.11 ^b^	22.40 ± 0.82 ^d^	455.82 ± 6.42 ^e^
	CB6:10	245.48 ± 6.58 ^d^	1986.02 ± 47.20 ^a^	312.84 ± 9.22 ^a^	21.83 ± 1.04 ^e^	522.03 ± 3.20 ^b^
	CB6A3	313.14 ± 6.43 ^b^	1967.03 ± 27.90 ^a^	306.04 ± 24.01 ^a,b^	26.04 ± 1.22 ^c^	459.72 ± 2.70 ^e^
	CB6A5	275.62 ± 9.18 ^c^	1814.01 ± 70.11 ^b^	304.63 ± 12.51 ^a,b^	26.92 ± 1.41 ^c^	490.71 ± 4.58 ^c^
Zn	CS	361.61 ± 7.98 ^a^	615.30 ± 32.60 ^c,d^	622.71 ± 14.70 ^b^	9.31 ± 0.62 ^h^	591.30 ± 1.65 ^g^
	CB4:3	330.40 ± 13.81 ^b^	630.01± 23.60 ^c^	546.30 ± 23.91 ^c^	20.12 ± 0.51 ^d^	620.01 ± 8.57 ^f^
	CB4:5	300.57 ± 18.70 ^d^	730.70 ± 82.40 ^a,b,c^	521.02 ± 29.91 ^c^	19.14 ± 0.82 ^de^	682.92 ± 1.57 ^d^
	CB4:10	291.01 ± 10.58 ^d^	746.50 ± 6.40 ^a^	529.42 ± 20.64 ^c^	58.12 ± 0.71 ^a^	789.63 ± 7.20 ^a^
	CB4A3	311.87 ± 18.15 ^cd^	719.30 ± 24.10 ^a,b^	531.03 ± 27.03 ^c^	48.73 ± 0.92 ^c^	730.31 ± 5.89 ^b^
	CB4A5	248.69 ± 18.21 ^e^	664.01 ± 36.30 ^b,c^	480.71 ± 12.45 ^d^	51.50 ± 0.41 ^b^	735.52 ± 8.20 ^b^
	CB6:3	361.04 ± 17.16 ^a^	705.70 ± 17.60 ^b^	539.32 ± 16.93 ^c^	21.14 ± 0.93 ^d^	697.74 ± 5.04 ^c^
	CB6:5	298.06 ± 16.39 ^d^	731.71 ± 27.01 ^a^	579.44 ± 23.92 ^c^	19.53 ± 0.44 ^d,e^	580.44 ± 8.76 ^g^
	CB6:10	300.58 ± 11.71 ^d^	689.70 ± 15.52 ^b^	628.53 ± 27.14 ^b^	17.82 ± 0.54 ^f^	663.73 ± 4.75 ^e^
	CB6A3	340.98 ± 17.73 ^a,b^	743.32 ± 54.80 ^a,b^	660.23 ± 11.12 ^a^	15.61 ± 0.33 ^g^	511.82 ± 9.32 ^h^
	CB6A5	329.03 ± 14.07 ^b,c^	664.03 ± 5.32 ^c^	613.52 ± 15.51 ^b^	18.42± 0.80 ^e,f^	677.63 ± 5.58 ^d,e^

Values = mean ± standard deviation (*n* = 3); Different lowercase letters (a,b,c,d,e,f,g,h) in the same column show significant differences (*p* < 0.05) between treatments.

**Table 4 molecules-28-02225-t004:** Component matrix of the data of the chemical fractions of Pb and Zn with other factors (OC, pH, EC) in the studied samples after a 30-day incubation.

Metal	Element	Component 1	Component 2
Pb	pH	0.42	−0.27
	OC	0.43	0.19
	EC	0.45	0.14
	F1	−0.42	−0.26
	F2	−0.24	0.52
	F3	0.23	−0.55
	F4	0.18	0.47
	F5	0.35	0.08
	Eigenvalue	4.67	1.51
	Cumulative variances (%)	58.41	77.33
Zn	pH	0.28	0.50
	OC	0.42	0.32
	EC	0.44	0.30
	F1	−0.41	0.12
	F2	0.23	0.10
	F3	−0.23	0.54
	F4	0.37	−0.37
	F5	0.37	−0.32
	Eigenvalue	3.93	2.31
	Cumulative variances (%)	49.18	78.08

**Table 5 molecules-28-02225-t005:** Designation of the incubation experiment.

Sample Plot	Sample Code	Biochar Weight (g)	Apatite Weight (g)	Soil Weight (g)	Ratio (%)
Control soil	CS	0.0	0.0	100	0
Control soil + 3% CB400	CB4:3	3.0	0.0	97	3
Control soil + 5% CB400	CB4:5	5.0	0.0	95	5
Control soil + 10% CB400	CB4:10	10.0	0.0	90	10
Control soil + 3% CB400 + 3% AP	CB4A3	3.0	3.0	94	3:3
Control soil + 5% CB400 + 5% AP	CB4A5	5.0	5.0	90	5:5
Control soil + 3% CB600	CB6:3	3.0	0.0	97	3
Control soil + 5% CB600	CB6:5	5.0	0.0	95	5
Control soil + 10% CB600	CB6:10	10.0	0.0	90	10
Control soil + 3% CB600 + 3% AP	CB6A3	3.0	3.0	94	3:3
Control soil + 5% CB600 + 5% AP	CB6A5	5.0	5.0	90	5:5

CB400: Corn cob-derived biochar produced at 400 °C; CB600: Corn cob-derived biochar produced at 600 °C; AP: apatite; CS: control soil; incubation time: 30 days.

## Data Availability

Not applicable.

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
