# Peer review of "Insight into the Speciation of Heavy Metals in the Contaminated Soil Incubated with Corn Cob-Derived Biochar and Apatite"

_molecules, 2023, doi:10.3390/molecules28052225_

Round 1
Reviewer 1 Report
The manuscript "Insight into the Speciation of Heavy Metals in the Contaminated Soil Incubated with Corn-Cob-Derived Biochar and Apatite" has been reviewed
Abstract is a sample and doesn't show the novelty.
Please show the hypothesis at the end of introduction.
The discussion is unsuitable to publish, you must focus on your work by discussing your results step by step and some of citations remove them from discussion is suitable to mention in section of introduction.
The conclusion need to rewrite again.
Author Response
First of all, the authors highly appreciate all valuable comments from the Reviewers for this manuscript. Each point has been scrutinized, revised and provided necessary information in the file attached below.

Reviewer 2 Report
The manuscript with titled “Insight into the Speciation of Heavy Metals in the Contaminated Soil Incubated with Corn-Cob-Derived Biochar and Apatite” was completely revised. It studies the treatment of contaminated soil at different application rates by using biochar and apatite. The physicochemical properties was investigated through FT-IR, SEM-EDS, and BET, and the removal of heavy metal was studied after 30 days of incubation. The manuscript is well written with promising data which I think it is interesting to broad of audience. However, a major revision is needed before being considered acceptable for publication.
Comments
1- Abstract should provide more details about the method and results
2- L150, peak at 1619 cm-1 is a very poor that was unnoticeable and should revised for its relation to the OH vibration.
3- The authors should make optimization of removal of metal ions firstly by studies different parameters; as contact time, temperature and material dose..etc.
4- L275 give a suitable reason for using rate 3:3 and 5:5 %
5- L558 please provide the country for the XRD and SEM
Author Response

(The authors gave the same response as above.)

Reviewer 3 Report
In general, the study achieved some good application results. However, there are a few points that need to be improved.
1. The author should add examples of immobilizing heavy metals in multiple-contaminated soil by CB400, CB600, and AP 32.
2. The author should illustrate and explain the mechanism of this application process.
3. Material chemistry, author should add some data such as TEM..., if possible.
Author Response

(The authors gave the same response as above.)

Reviewer 4 Report
This work studies the response of a soil to some trace elements, after the addition of residues (corn biochar and apatite) that, according the authors “alter soil properties”. The investigation is the continuation of very similar previous works of the components of the team. This reviewer here exposes some reasons to reject the manuscript in the actual form. After addressing this new perspective of the content, the manuscript presents some format requirements, some minor spelling mistakes, but an overly long and repetitive “results and discussion” heading.
line 28 residue check spelling
1. INTRODUCTION
Although the background is enough, the aims only take into account 2 elements among the common trace elements. These kinds of works should present data for the main group of trace metals that are usually supplied to soils from human activities (Cd-Cr-Cu-Ni-Pb-Zn, and even As). The study of only part of the elements diminishes the quality of the work. You are developing sequential extraction and measuring by ICP, so you should apply for a a large set of elements because, even if they are seemed to be present in "normal" concentrations, they can present a mobile percent higher than the expected.
2. RESULTS
METHODOLOGY should be placed before results!
Line 86: define the acidity of soil according the Ph scale ("somewhat" seems to be too colloquial). FAO´s classification indicates "slightly acid" which is a very low acidity.
Line 95 Check and unify the decimal separator (check the whole document)
Line 98. you are searching a remediation for a soil, but not for only one "soil sample"
FT-IR analysis of amendments: Here the authors expose a very long explanation about the employ of the technique FT-IR in the amendments, but I cannot find the usefulness in this experience. After this determination, you don´t use the data form the discussion. It is not connected to the rest of the paper. Moreover I can find this characterization in the previous studies you cite in the bibliography. I cannot fit this paragraph with the rest of the body. In my opinion, this paragraph does not add information to the manuscript, and yet it implies the mention of a large number of bibliographic citations; so I suggest its suppression here, or a clear use of this information to help the discussion and conclusions.
Figure 2 B and D. The length of the scales is not clear, is it from the first vertical line to the end of the image at the right?
Line 178. Avoid repetition: Figures 3 (A, B, C, D). Figures 3A, B, C, D
Also, how can you explain such change in structure among CB400 and CB600?
Line 183-186 …”that the functional groups 183 on the surface of CB400 and AP might facilitate the immobilisation of heavy metals 184 through precipitation or exchange reactions, while CB600 might facilitate the adsorption 185 of heavy metals due to its large specific surface area” These are suppositions, since the mechanisms were not studied (even not the minerals involved to select the probable mechanism). It is not enough for this kind of studies.
Lines 190-210 " Alteration of OC, pH, and EC after a 30-day incubation with biochar and apatite"
This paragraph reflects only the fact that, lesser the proportion of soil and higher the amendment, lesser the content in Zn and Pb. It does not reflect the matter of time, because different proportion of mixtures were made. They are results that cannot be related with time.
Lines 233-on
A sequential study should be addressed as the part of a complete especiation study. But if you do not determine which phases (minerals) are really involved in each step, the procedure loses its significance, since you cannot prove the coherency of chamical data with real compounds containing such elements and then, all results are based on assumptions
MATERIALS AND METHODS (in addition of putting into place)
A location map should be provided, with the place of the samples.
Line 527 "All incubated soil samples were placed in the shadow at 527 ambient temperature for 30 days". If you are searching a low cost treatment for soil amendment, I cannot share this first step. The "incubation" process would limit the applicability of the amendment if you try to remediate high volumes of soils. Is it absolute necessary? In this case, you should justify it as well as exposing the disadvantages.
Line 528. Another drawback is the addition of distilled water to simulate de humidity, which is far from real since precipitation water is ion charged.
Lines 546-548. Repeated terms in the phrase.
Line 553. Correct: were 92.11 ÷ 109.27%
Lines 558-564. You mention XRD but you didn´t show any mineralogical diagram, please include in the results a type diagram of the main mineralogy of the soils submitted to the sequential extraction in order to see the minerals involved in metal retention (Sequential extraction by itself cannot be used to support discussion on it, since you cannot state where the chemical compounds are contained)
4.CONCLUSIONS
Line 576. This first state is not a conclusion, Clear it.
Line 585. “The biochar and apatite could immobilize Pb and Zn in the untreated soil via cation exchange, precipitation reaction, complexation, and adsorption” This phrase is not clear. How can the amendments help to immobilice metals if the spoil is untreated? Please explain.
Line 587. “At the same time, the amendments also could increase the F2, F3, F4, and F5 of Pb and Zn 587 by turning them into more stable forms in natural conditions.” Here you made a very important state, but you did not give sense to it. You refer to chemical data to mention that this occurs, but you don´t give any explanation about how it is possible. Do you think that in 30 days soy have formed carbonates, Fe o Mn oxides and residual minerals, and these retain Pb and Zn??
Author Response
First of all, the authors highly appreciate all valuable comments from the Reviewers for this manuscript. Each point has been scrutinized, revised and provided necessary information as below.

Round 2
Reviewer 1 Report
I think this manuscript can be published in the present form
Reviewer 2 Report
After check the revised version of manuscript titled “insight into the speciation of heavy metals in the contaminated soil incubated with corn-cob-derived biochar and apatite”. I see the manuscript was improved and it is suitable for publication in present form.
Reviewer 4 Report
The authors have addressed all the suggestions. The manuscript can be accepted in the present form.